# Efficient Neural Architecture Search on Low-Dimensional Data for OCT Image Segmentation

**Nils Gessert**[1]                     nils.gessert@tuhh.de

**Alexander Schlaefer**[1]                  schlaefer@tuhh.de

[1]*Institute of Medical Technology, Hamburg University of Technology, Germany*

## Abstract

Typically, deep learning architectures are handcrafted for their respective learning problem. As an alternative, neural architecture search (NAS) has been proposed where the architecture's structure is learned in an additional optimization step. For the medical imaging domain, this approach is very promising as there are diverse problems and imaging modalities that require architecture design. However, NAS is very time-consuming and medical learning problems often involve high-dimensional data with high computational requirements. We propose an efficient approach for NAS in the context of medical, image-based deep learning problems by searching for architectures on low-dimensional data which are subsequently transferred to high-dimensional data. For OCT-based layer segmentation, we demonstrate that a search on 1D data reduces search time by 87.5 % compared to a search on 2D data while the final 2D models achieve similar performance.

**Keywords:** Neural Architecture Search, Deep Learning, Segmentation, OCT

## 1. Introduction

Over the last years, manual feature engineering has been replaced by deep learning approaches such as convolutional neural networks (CNNs) for numerous medical, image-based learning problems (Litjens et al., 2017). CNNs itself are often difficult to design and it is unclear what kind of architecture is suitable for which learning problem. Therefore, *neural architecture search* (NAS) has been proposed. Typical NAS approaches include grid search, genetic algorithms, bayesian optimization or random search (Kandasamy et al., 2018). Recently, reinforcement learning (RL) methods have been proposed where a recurrent controller is trained to predict an architecture's structure by maximizing the architecture's expected validation performance as a reward (Zoph and Le, 2016). This approach has been successful for 2D image classification problems (Liu et al., 2018; Zoph et al., 2018).

The concept of NAS is also very promising for the medical image domain as there is a vast amount of imaging modalities and learning problems that require architecture design. However, NAS can be very time-consuming which is even more problematic for medical image data which is often 3D or 4D in nature (Li et al., 2008). Some approaches have used lower dimensional data representations such as 2D slices instead of full 3D volumes in order to reduce computational effort (Litjens et al., 2017). However, many approaches have shown that considering higher dimensional context can improve performance (Kamnitsas et al., 2017; Gessert et al., 2018a,b).

We propose an efficient NAS approach for segmentation with mutlidimensional medical image data. To overcome long architecture search times, we perform the search on lower dimensional data which leads to shorter search times. Then, we transfer the learned architecture to the higher, target dimension. We show the concept for the example task of retinal layer segmentation with optical coherence tomography (OCT) data as the problem can be addressed in 1D (A-Scan segmentation) and 2D (B-Scan segmentation). Adopting the efficient neural architecture search (ENAS) framework (Pham et al., 2018), we learn submodules for a U-Net-like (Ronneberger et al., 2015) architecture. We demonstrate that our learned architecture outperforms a ResNet-inspired (He et al., 2016) baseline and that an architecture learned on 1D data transfers well to 2D data.

## 2. Methods

**Dataset.** We use a publicly available OCT dataset with images from patients with mild age-related macular degeneration (AMD) and normal subjects (Farsiu et al., 2014). Experts provided layer boundaries for the inner limiting membrane (ILM), retinal pigment epithelium drusen complex (RPEDC) and Bruchs membrane (BM). We generate pixel-wise annotations by assigning classes to tissue layers in between boundaries, i.e., ILM to RPEDC is class 1, RPEDC to BM is class 2 and BM to the end is class 3. The image space above the ILM is treated as background. Note that directly learning the boundaries can be beneficial for this problem (Roy et al., 2017). We chose a pixel-wise encoding to have a representative medical segmentation task that can be addressed with a standard U-Net.

**Baseline Model.** As a baseline we use a U-Net-like model. The model takes a 1D A-Scan or a 2D B-Scan as its input and predicts a segmentation map with the same size as the input. For the long-range connections we use summation, following (Yu et al., 2017). We use ResNet blocks in the network. Convolutions use a kernel size of 3 and extensions from 1D to 2D are performed by extending all kernels isotropically by an additional dimension.

**ENAS U-Net.** Next, we adopt the ENAS framework (Pham et al., 2018) for image classification to image segmentation with a U-Net. To simplify the architecture search space, we keep the general U-Net structure fixed and only learn new module blocks, similar to the micro search space in ENAS. The input/output and downsampling/upsampling layers also stay fixed. For the module search space, we let the controller learn the properties of 2 cells each containing 2 subcells. The cells' output is the summation of the subcells' output. For each subcell, the controller defines its input (the module input or another cell's output) and its operation. Similar to ENAS, we allow five basic operations for the controller to choose from: convolutions with kernel size 3 or 5, average- and max-pooling with kernel size 3 and the identity transform.

**Training and Evaluation.** We consider a training set of 150 volumes (model training), a reward set of 56 volumes (controller training), a validation set of 2 volumes and a test set of 60 volumes. We follow ENAS with interleaved training of the model (dice loss) and the controller (dice score reward). After training for 200 epochs, we sample 20 architecture configurations from the controller and evaluate them on the validation set. Then, we select the best-performing configuration and retrain the model from scratch on the training set. Finally, we evaluate the model's performance on the test set. For the baseline model, we train on the training set for 200 epochs and evaluate on the test set afterwards.

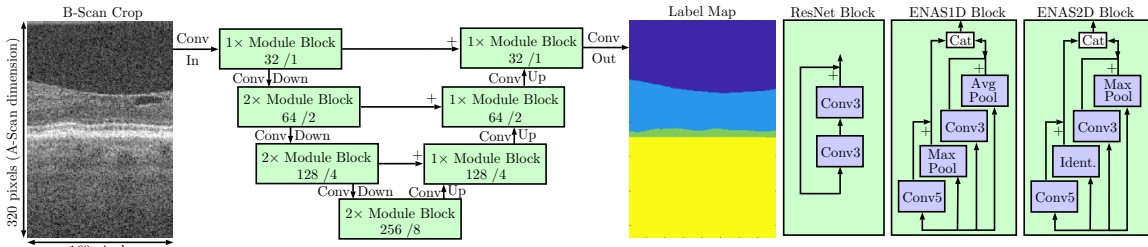

Figure 1: Left, the architecture with a 2D B-Scan input is shown. In each block the number of feature maps and total stride is given. Right, the baseline ResNet block and two learned ENAS blocks are shown.

Table 1: Dice and search time, performed on an NVIDIA GTX 1080 Ti.

| Model | Mean Dice | Search Time |
|---|---|---|
| ResNet U-Net 1D | $0.916 \pm 0.10$ | — |
| ENAS1D U-Net 1D | $\mathbf{0.928 \pm 0.09}$ | 1 h 30 min |
| ResNet U-Net 2D | $0.962 \pm 0.05$ | — |
| ENAS1D U-Net 2D | $0.971 \pm 0.04$ | 1 h 30 min |
| ENAS2D U-Net 2D | $\mathbf{0.973 \pm 0.04}$ | 12 h 0 min |

## 3. Results and Discussion

The architecture and the learned modules are shown in Figure 1. The results are shown in Table 1. Both the 1D and 2D architectures learned with ENAS on 1D data outperform the ResNet baseline. Notably, the increase is achieved without altering fundamental and potentially more impactful U-Net properties such as the encoder-decoder structure or the long-range connections. As a next step, these properties could be included in the search space which was successful for segmentation in the natural image domain with DeepLab-based architectures (Liu et al., 2019).

Performing a search on 1D data substantially decreases the search time by 87.5 % compared to a search on 2D data while performance differences are marginal. This is particularly interesting as the OCT data is not isotropic and the spatial dimensions are quite different. This indicates that learning on low-dimensional, less resource demanding data representations is a viable approach for NAS. Thus, extension to other problems such as brain segmentation might be feasible, e.g., by performing NAS on axial slices before applying the discovered architectures on 3D volume data.

Summarized, we propose an efficient approach for NAS in the context of multidimensional medical image data. We demonstrate that searching for an architecture on low-dimensional data transfers well to high-dimensional data. An architecture discovered on 1D data performs similar to one discovered on 2D data while substantially reducing search time. Our approach could enable efficient NAS for a variety of medical learning problems.

## Acknowledgments

This work was partially funded by the TUHH $I^3$-Labs initiative.

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
