# OpenReview forum: "Efficient Neural Architecture Search on Low-Dimensional Data for OCT Image Segmentation"
_MIDL.io/2019/Conference/Abstract — MIDL Abstract 2019_

### Official Review · AnonReviewer2 · 2019-04-29
**Abstract Review**

**Rating:** 4
**Confidence:** 3

**Review:**

The authors proposed an alternative to neural architecture search (NAS)  for architectures on low-dimensional data which are subsequently transferred to high-dimensional data. They report a %85.7 acceleration using a search on 1D data for OCT-layer segmentation instead of 2D models.

Pros

- The issues of optimal network architecture is an an often overlooked-issue in medical image analysis and the authors perform a comprehensive analysis for an abstract.
- The proposed NAS technique is compared with its variants and showed promising results.
- The abstract is well-written and the method is well described.

Minor suggestions
- Exhaustive analysis of neural architecture search techniques on various medical image analysis tasks is necessary to make bigger claims (e.g. allowing a search on multiple layers and parameters, different medical image analysis challenges)

---

### Official Review · AnonReviewer1 · 2019-05-01
**too few details and the experiments are not well explained**

**Rating:** 2
**Confidence:** 2

**Review:**

There are too few details on the proposed NAS approach and why it should be used for OCT segmentation. The motivation is too generic but not well suited for the problem. No better experimental results are observed.

---

### Decision · Program_Chairs · 2019-05-06
**Acceptance Decision**

Accept